DISCOVERY REPORT

# Multidrug-resistant *E. coli* encoding high genetic diversity in carbohydrate metabolism genes displace commensal *E. coli* from the intestinal tract

Christopher H. Connor[1,2], Amanda Z. Zucoloto[2,3,4], John T. Munnoch[5], Ian-Ling Yu[3], Jukka Corander[6,7,8], Paul A. Hoskisson[5], Braedon McDonald[2,3,4]*, Alan McNally[1]*

1 Institute of Microbiology and Infection, College of Medical and Dental Science, University of Birmingham, Birmingham, United Kingdom, 2 International Microbiome Centre, University of Calgary, Calgary, Canada, 3 Department of Critical Care Medicine, Cumming School of Medicine, University of Calgary, Calgary, Canada, 4 Calvin, Phoebe, and Joan Snyder Institute for Chronic Diseases, Cumming School of Medicine, University of Calgary, Calgary, Canada, 5 Strathclyde Institute of Pharmaceutical and Biomedical Science, University of Strathclyde, Glasgow, United Kingdom, 6 Department of Biostatistics, Institute of Basic Medical Sciences, University of Oslo, Oslo, Norway, 7 Parasites and Microbes, Wellcome Sanger Institute, Cambridge, United Kingdom, 8 Helsinki Institute of Information Technology, Department of Mathematics and Statistics, University of Helsinki, Helsinki, Finland

* bamcdona@ucalgary.ca (BM); a.mcnally.1@bham.ac.uk (AM)

## Abstract

Extra-intestinal pathogenic *Escherichia coli* (ExPEC) can cause a variety of infections outside of the intestine and are a major causative agent of urinary tract infections. Treatment of these infections is increasingly frustrated by antimicrobial resistance (AMR) diminishing the number of effective therapies available to clinicians. Incidence of multidrug resistance (MDR) is not uniform across the phylogenetic spectrum of *E. coli*. Instead, AMR is concentrated in select lineages, such as ST131, which are MDR pandemic clones that have spread AMR globally. Using a gnotobiotic mouse model, we demonstrate that an MDR *E. coli* ST131 is capable of out-competing and displacing non-MDR *E. coli* from the gut in vivo. This is achieved in the absence of antibiotic treatment mediating a selective advantage. In mice colonised with non-MDR *E. coli* strains, challenge with MDR *E. coli* either by oral gavage or co-housing with MDR *E. coli* colonised mice results in displacement and dominant intestinal colonisation by MDR *E. coli* ST131. To investigate the genetic basis of this superior gut colonisation ability by MDR *E. coli*, we assayed the metabolic capabilities of our strains using a Biolog phenotypic microarray revealing altered carbon metabolism. Functional pangenomic analysis of 19,571 *E. coli* genomes revealed that carriage of AMR genes is associated with increased diversity in carbohydrate metabolism genes. The data presented here demonstrate that independent of antibiotic selective pressures, MDR *E. coli* display a competitive advantage to colonise the mammalian gut and points to a vital role of metabolism in the evolution and success of MDR lineages of *E. coli* via carriage and spread.

**Data Availability Statement:** All relevant data are within the paper and its Supporting Information files. All genomic data used in this study can be found in a dedicated figshare repository 10.6084/m9.figshare.c.6147189.

**Funding:** This work was funded by a Wellcome Trust funded MIDAS PhD studentship awarded to CC (Grant number 203821/Z/16/A). The funders had no role in study design, data collection and analysis, decision to publish, or preparation of the manuscript.

**Competing interests:** The authors have declared that no competing interests exist.

**Abbreviations:** AMR, antimicrobial resistance; CFU, colony-forming unit; ESBL, extended spectrum beta-lactamase; ExPEC, extra intestinal pathogenic *Escherichia coli*; IDT, Integrated DNA Technologies; MDR, multidrug resistance; SNP, single nucleotide polymorphism; ST, sequence type.

# Introduction

Infections by multidrug resistant (MDR) gram-negative pathogens now represent one of the greatest global public health challenges of our generation, with the World Health Organisation declaring them of utmost international importance. Chief among these pathogens is MDR *Escherichia coli*, which are responsible for an alarming rise in the incidence of antimicrobial resistant (AMR) blood stream and urinary tract infections [1]. MDR in *E. coli* is heavily associated with the carriage of large MDR plasmids encoding extended spectrum beta-lactamases (ESBLs) and carbapenemases such as NDM, KPC, and Oxa-48 that confer resistance to third-generation cephalosporins and carbapenem classes of antibiotics, respectively [1]. Intriguingly, such plasmids are very rarely found in intestinal pathogenic *E. coli* such as *E. coli* O157 or common enteropathogenic and enterotoxigenic *E. coli* strains. Rather MDR plasmid carriage is concentrated in a number of lineages responsible for extra-intestinal pathogenesis such as blood stream and urinary tract infections [2].

Extra-intestinal pathogenic *E. coli* (ExPEC) is the name given to *E. coli* strains capable of causing extra-intestinal infections, but do not represent a phylogenetically distinct group of organisms. Rather ExPEC are found across the species phylogeny mainly in phylogroups B2, D, and F [3]. Recent longitudinal surveys of national blood stream infection isolates have shown the most common ExPEC lineages to be ST131, ST73, ST69, ST95, ST410, and members of the ST10 complex including ST167 [4,5]. Equally as intriguingly, MDR plasmids are not evenly distributed among these ExPEC lineages but rather their carriage is concentrated in a small number of highly successful, globally disseminated clones [2]. The most successful of these clones is clade C of ST131, the most common cause of MDR blood stream and urinary tract infections worldwide [6]. Other common MDR *E. coli* lineages include ST69, ST410, ST167, and ST648 [2].

What makes certain clones or lineages of *E. coli* successful MDR pathogens is an ongoing question. Recent analysis of longitudinal blood stream infection isolates from Norway shows that ST131 strains successfully emerged to be dominant in the absence of MDR plasmid carriage indicating MDR alone is not the driver of their success [5]. Analysis of longitudinal UK isolates shows that MDR alone is not sufficient to drive strains to complete dominance of the epidemiological landscape [4]. Recent evidence suggests an important phenotype that differentiates MDR *E. coli* from other lineages is their ability to rapidly and asymptomatically colonise the intestinal tract of humans. The COMBAT study of 1,847 people travelling from the Netherlands to Asia, Africa, and South America found that 34% of those travelling acquired an ESBL *E. coli* in their intestinal tract during their journey, with that number increasing to 75% of those travelling to Asia [7]. Of those colonised in the study, 11% were colonised for up to 12 months. A small scale study of University of Birmingham students found that all participants travelling to Asia became colonised by an ESBL *E. coli*, with genomic analysis confirming that this was due to acquisition of a new MDR strain and not the resident commensal *E. coli* becoming MDR [8]. A recent study of medical personnel travelling to Laos sampled travellers in real time to show that every single person was colonised by an MDR *E. coli* during travel, with colonisation occurring within days after arrival [9]. A very recent study by our group deploying metagenomic sequencing on the COMBAT study samples showed that when people were colonised by an MDR *E. coli*, there was no detectable impact on diversity or composition of the wider gut microbiome as a result of MDR colonisation [10]. While observational studies have yielded hypothesis-generating data suggesting stain-to-strain competition, this has never been directly tested in vivo.

Studies investigating genetic determinants differentiating MDR *E. coli* lineages from the rest of the population have also uncovered a number of parallel observations. Studies

comparing clade C ST131 to its ancestral population show a collection of adaptive nucleotide substitutions in chromosomal promoter regions associated with the specific plasmid carried by the strain [11], a pattern which has also been seen in ST167 [12]. A high-resolution study of ST131 using a pangenome approach to identify allelic variations in genes found a highly elevated number of nucleotide substitutions in genes involved in mammalian colonisation including anaerobic metabolism, iron acquisition, and adhesins. This pattern was not seen in successful non-MDR ExPEC lineages such as ST73 [13]. Such allelic diversity in anaerobic metabolism genes has also been seen in ST167 and ST648 [12,14], and it was also shown that recombination of new alleles of *fhu* iron acquisition genes was the key evolutionary event underpinning the emergence the carbapenem-resistant B4/H24RxC clone in the ST410 lineage [15]. We hypothesise that these genetic adaptations may contribute to more effective colonisation of the mammalian gut.

Here, we use a gnotobiotic mouse model of intestinal colonisation to directly test the hypothesis that MDR *E. coli* can outcompete commensal *E. coli* via inter-strain competition to establish dominant colonisation of the intestinal tract in vivo in the absence of any antibiotic selection. We demonstrate that an MDR ST131 strain can out-compete a commensal strain to establish intestinal colonisation in gnotobiotic mice. Furthermore, when introduced into the gut of mice pre-colonised with commensal *E. coli*, MDR ST131 could displace commensal *E. coli* from the gut to establish dominant colonisation. Displacement of the resident commensal strain occurs within 48 h of cohousing with mice colonised by MDR ST131, with ST131 becoming the dominant strain in all co-housed mice. To understand the biological process underpinning this competition, we use Biolog phenotypic microarray which identifies altered utilisation of carbon sources. Targeted genomic comparison our assayed strains reveals distinct mutational signatures. We further expand this analysis to a dataset of 19,571 genomes representing the full phylogenetic and AMR diversity of *E. coli* to identify a significant link between metabolism and carriage of AMR genes. We find that MDR lineages of *E. coli* display an increased nucleotide diversity in genes associated with carbohydrate utilisation which may afford competitive advantage for colonising the mammalian gut compared to commensal *E. coli* strains.

## Results

### Non-MDR ExPEC and MDR ExPEC are efficient colonisers of germ-free mice, with both ExPEC outcompeting a commensal isolate in co-colonisation

To determine whether MDR *E. coli* is able to out-compete non-MDR *E. coli* in the intestine in vivo, we performed competitive colonisation experiments in germ-free mice using 3 different strains of *E. coli*: a non-MDR (ST73) commensal strain 822-E8 isolated from a healthy human volunteer, a pathogenic non-MDR ExPEC (ST73) strain F084 isolated from a bacteraemia patient, and a pathogenic MDR ExPEC (ST131) strain F016 isolated from a bacteraemia patient (S3 Table). All 3 strains possessed an equivalent virulence-associated gene profile; however, the MDR strain F016 possessed a greater number of AMR genes (S1 and S2 Figs). Mice were inoculated with $10^9$ colony-forming unit (CFU) of each strain via oral gavage and bacterial growth was measured by enumeration of CFU from the faeces as well as strain specific qPCR. Under these conditions, all strains could individually monocolonise germ-free mice (S3 Fig). Competitive colonisation between strains was investigated by orally gavaging GF mice with a 1:1 ratio of 2 strains in combination (see Fig 1A for combinations) followed by quantification of each strain in the faeces over the subsequent week (Fig 1B–1E). Both F084 and F016 strains out-competed the commensal 822-E8 strain achieving 96.1% and 80.7% of the total

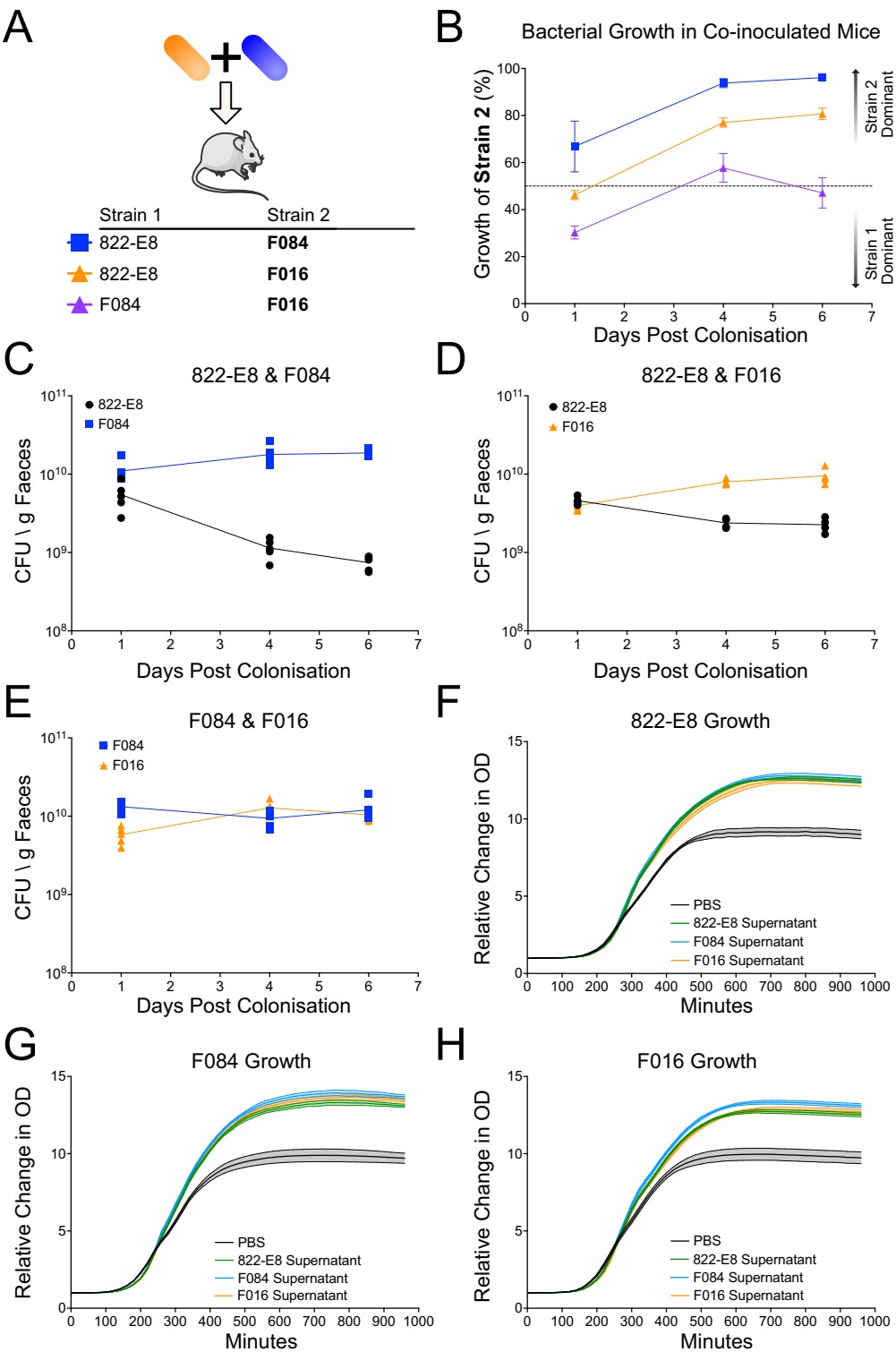

**Fig 1. Co-colonisation of mice with 2 strains of *E. coli* and supernatant growth curves.** (A) Schematic of mouse colonisation set up, 2 strains of *E. coli* are mixed immediately prior to gavage. (B) Growth of *E. coli* in co-colonised mice measured by strain specific qPCR. Growth of the strain 2 listed in the schematic is plotted as a percentage of total growth against days post gavage ($n = 5$, 822-E8 and F084, growth of F084 in blue; $n = 4$, 822-E8 and F016, growth of F016 in orange, $n = 5$ F084 and F016, growth of F016 in purple). Dotted line at 50% indicating equivalent growth of both strains,

values higher than 50% indicate higher growth of strain 2, while values less than 50% indicate lower growth of strain 2. (C–E) CFU per gram of faeces as measured by qPCR for mice colonised simultaneously by 822-E8 (black) and F084 (blue) (C), 822-E8 (black) and F016 (orange) (D), or F084 (blue) and F016 (orange) (E). (F–H) Growth of 822-E8 (F), F084 (G), or F016 (H) strain in 50% LB + 50% PBS (black), 822-E8 supernatant (green), F084 supernatant (blue), or F016 supernatant (orange), (*n* = 6). Parts of the figure were drawn by using pictures from Servier Medical Art. Servier Medical Art by Servier is licensed under a Creative Commons Attribution 3.0 Unported License (https://creativecommons.org/licenses/by/3.0/). Raw data used to produce figures available in S1 and S2 Data.

growth, respectively, by day 6 post gavage (Fig 1B–1D). Neither F084 nor the MDR F016 was able to out-compete the other with F016 accounting for 47.1% of the growth by day 6 (Fig 1B and 1E).

It is possible that F084 and MDR F016 are out-competing the commensal 822-E8 strain due to phage or production of a secreted toxin. Previous data from our group has shown that an ST131 strain retards growth of both ST73 and ST10 strains in LB broth [16]. Therefore, we grew each strain in LB broth overnight, filter sterilised the spent medium before mixing in a 1:1 ratio with fresh LB, and investigated the growth kinetics of all strains in the presence of supernatant from a competing strain. No strain displayed any impairment in growth in the presence of either autologous or heterologous supernatant indicating that no strains were releasing toxins/phage to kill competing strain (Fig 1F–1H).

## MDR ExPEC efficiently displaces an established commensal from the mouse intestinal tract

Next, we aimed to determine whether MDR *E. coli* could displace commensal *E. coli* that had already established a colonisation niche within the intestine. Germ-free mice were colonised with a commensal strain 822-E8 by oral gavage and colonisation allowed to stabilise for 1 week before challenging with a second *E. coli* strain (Fig 2A). When commensal-colonised mice were challenged with MDR strain F016, it rapidly out-competes the commensal strain 822-E8 within 4 days accounting for greater than 60% of the CFU (Fig 2B and 2C). By day 21, the MDR strain F016 accounted for 80.4% of the CFU in the faeces. In contrast, this displacement is not observed when commensal-mice colonised are challenged with the non-MDR strain F084, which results in an equilibrium of 50–50 colonisation of both strains within 4 days and remains equivalently co-colonised at 21 days post-challenge (Fig 2B and 2D). Conversely, when MDR strain F016 monocolonised mice are challenged with commensal strain 822-E8, the commensal is unable to displace F016, accounting for 20.4% of the CFU at day 4 and further diminishing to 15.3% at day 21 (Fig 2B and 2E). These results are not due to changes in commensal colonisation as growth remains stable in the control group throughout the experiment (Fig 2F). Collectively, these data demonstrate that MDR *E. coli* strain F016 can displace the commensal *E. coli* strain 822-E8 to establish itself as a dominant coloniser of the gut in vivo.

To determine whether our findings using oral gavage (with large quantities of bacteria) could be replicated in the setting of environmental exposure to MDR *E. coli*, we monocolonised mice wither either commensal 822-E8 or MDR ExPEC F016. After allowing colonisation to stabilise for 7 days mice were co-housed (Fig 3A). Within 48 h, F016 is observed in the faeces of all mice and becomes the dominant coloniser in all mice by day 4 accounting for nearly 80% of the CFU (Fig 3B). This colonisation dominance is persistent and sustained for many weeks (Fig 3B). Of note, F016 monocolonised mice did acquire a low level of colonisation by 822-E8 following co-housing, but F016 remained the dominant strain with 822-E8 accounting for less than 20% of faecal CFUs. This MDR F016 phenotype is not due to strain-specific differences in inherent ability to colonise the mouse gut, as the total CFU recovered from F016

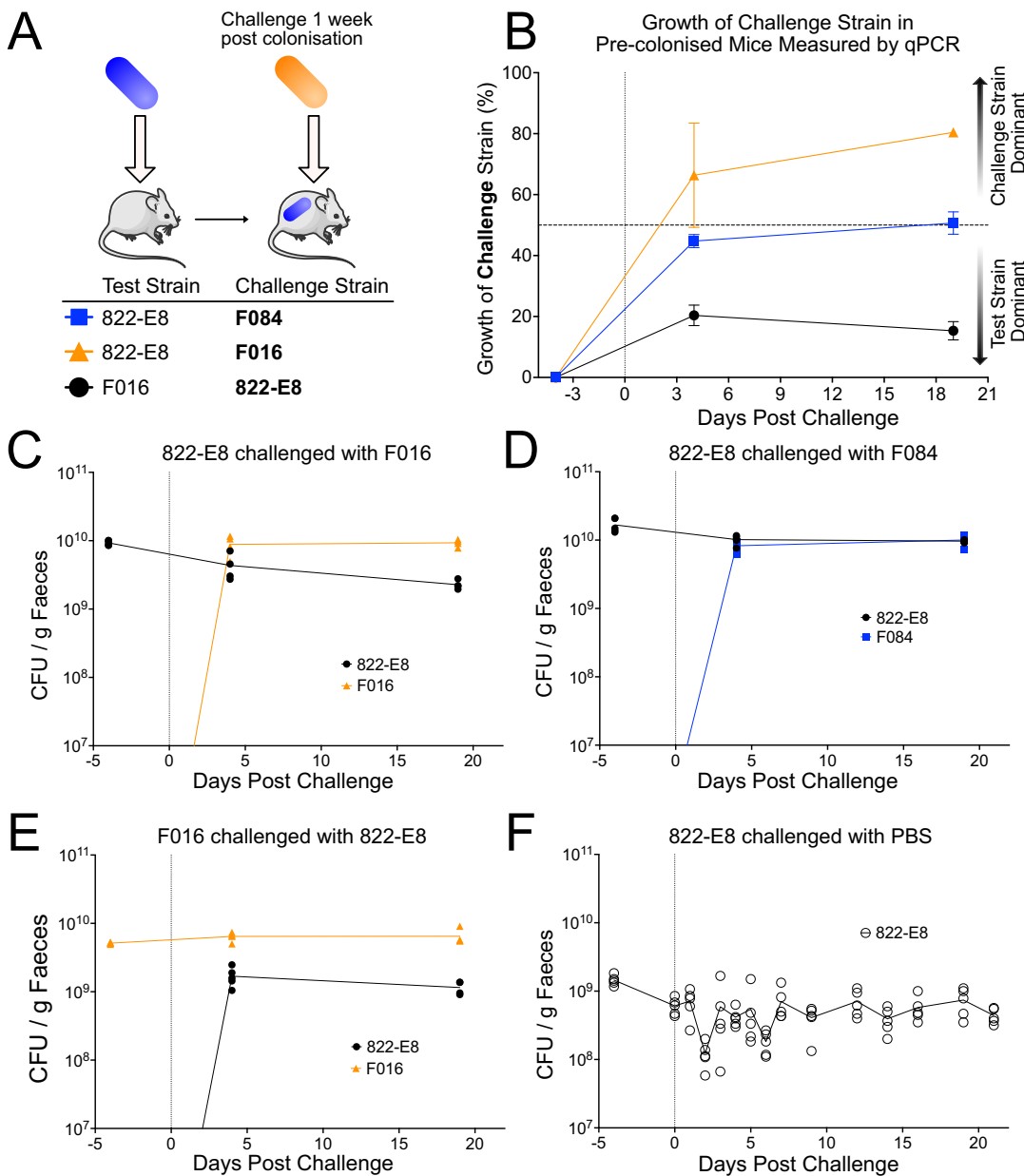

**Fig 2. Mice colonised with a test strain for 1 week before challenge with a second strain.** (A) Schematic of displacement experiment in which mice are pre-colonised with a $10^9$ CFU of the test strain followed a week later by a second gavage with $10^9$ CFU of a challenge strain. (B) Growth of the challenge strain described in panel A in pre-colonised mice measured by strain specific qPCR at select time points. Data presented as percentage of total growth attributable to challenge strain ($n = 5$ for 822-E8 vs. F084, growth of F084 in blue, $n = 4$ for 822-E8 vs. F016, growth of F016 in orange, $n = 4$ for F016 vs. 822-E8, growth of 822-E8 in black). Vertical dotted line indicates time point for challenge with second strain. Horizontal dashed line indicates 50% corresponding to equivalent colonisation of test and challenge strains. (C–E) Growth of both colonising strains in mice measured by strain-specific qPCR, CFU calculated against a standard curve and normalised to faecal pellet weight. Growth of 822-E8 strain in black circles, growth of F084 in blue squares and growth of F016 in orange triangles. Vertical dotted line indicates time of challenge with second strain. (F) Growth of 822-E8 in colonised mice challenged with PBS. Growth measured by CFU enumeration from plates at regular intervals. Vertical dotted line indicates time of challenge with PBS. Parts of the figure were drawn by using pictures from Servier Medical Art. Servier Medical Art by Servier is licensed under a Creative Commons Attribution 3.0 Unported License (https://creativecommons.org/licenses/by/3.0/). Raw data used to produce figure available in S3 Data.

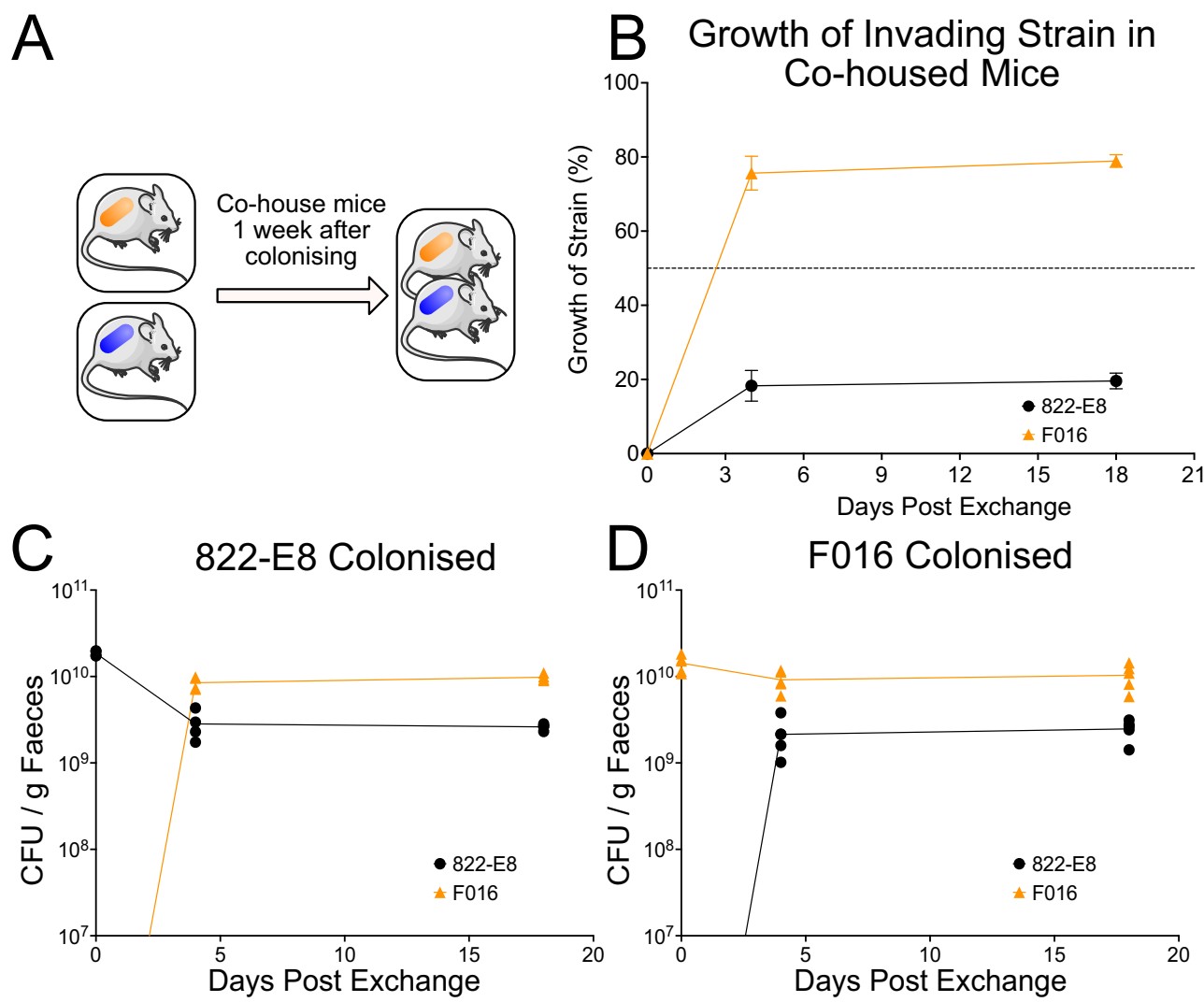

**Fig 3. Transmission of strains between monocolonised mice when co-housed.** (A) Schematic of co-housing experiment in which mice are monocolonised with either 822-E8 or F016 strains for 1 week before exchanging mice between cages. (B) Growth of the invading strain in co-housed mice measured by strain-specific qPCR at select time points ($n$ = 4 for 822-E8 shown in black, $n$ = 5 for F016 shown in orange). Data presented as a percentage of total growth. (C, D) CFU per gram of faeces measured by qPCR in monocolonised mice followed by co-housing. Bacterial growth measured by strain specific qPCR from faecal pellets, 822-E8 growth (black circles) and growth of F016 (orange triangles). (C) Mice monocolonised by an 822-E8 strain ($n$ = 4). (D) Mice monocolonised by F016 ($n$ = 5). Parts of the figure were drawn by using pictures from Servier Medical Art. Servier Medical Art by Servier is licensed under a Creative Commons Attribution 3.0 Unported License (https://creativecommons.org/licenses/by/3.0/). Raw data used to produce figure available in S4 Data.

monocolonised mice is equivalent to 822-E8 monocolonised mice (Fig 3C and 3D). The host response can influence the ability of certain pathogens to establish colonisation. The host immune response in the small intestine and colon of colonised mice was analysed histologically revealing no evidence of inflammatory cell recruitment or tissue damage in response to colonisation by F016 (S5 Fig). Expression of 11 cytokines was assayed by probe-based qPCR from tissue collected from the small intestine, caecum, and colon. Expression of the assayed cytokines in the small intestine and colon revealed few differences between colonisation conditions (S6–S8 Figs).

Given that our displacement findings cannot be driven by classical virulence genes (S1 Fig) nor differences in host response, we sought to perform a comprehensive comparative genomic and phenotypic analysis of the *E. coli* strains competed in vivo to identify the factors underpinning F016's colonisation success.

## The MDR ST131 strain F016 displays altered carbon source utilisation alongside numerous polymorphisms in metabolic genes

Stains F084, F016, and 822-E8 were subject to Biolog analysis using commercially available Phenotype MicroArray plates. Significant differences were only detected in the PM1 carbon utilisation plate. Of the 96 conditions tested in PM1, 34 showed statistical (<0.05 *P*-value) differences between the 3 strains (Fig 4A). The data shows that the MDR strain F016 is less efficient at utilising N-Acetyl-Glucosamine, trehalose, mannose, xylose, fructose, maltose, melibiose, methyl-D-galactoside, lactose, and lactulose than strains F084 and 822-E8. Conversely, strain F016 is far more efficient at utilising keto-butyric acid, sucrose, L-glutamine, hydroxy-butyric acid, D and L-threonine, and glyoxylic acid. Additionally, both F084 and F016 were significantly more efficient at utilising both propionic and mucic acid. To investigate the genomic factors underpinning these phenotypic differences, we examined genetic polymorphisms in 73 genes associated with the metabolites above. Using the commensal 822-E8 strain as a reference revealed that there was a very low number of polymorphisms in the F084 strain, while the F016 strain displayed a far higher number of mutations. The majority of mutations were single nucleotide polymorphisms (SNPs) that caused synonymous mutations or missense mutations (Fig 4E and 4F). In addition to mutational profiling, the strains were subject to a functional pangenomic analysis. A pangenome of the 3 strains was constructed, the resulting pangenome reference file was functionally annotated using the eggNOG database with the eMapper utility. This analysis identified 712 genes uniquely present and 466 genes uniquely absent in the F016 strain. The majority of these genes were annotated with "S -function unknown" (238/712 and 98/466) alongside a significant number with no functional annotation (106/712 and 60/466). Of the uniquely present genes, the most abundant COG categories were "Replication, recombination and repair," "Transcription," and "Carbohydrate metabolism" with 62, 50, and 37 genes, respectively. While the uniquely absent genes were abundant with "Replication, recombination and repair," "Cell membrane biogenesis," and "Transcription" with 40, 39, and 25 genes, respectively. Based on phenotypic differences, we focussed on the carbon metabolism genes that were differentially present in F016 revealing that it possessed multiple genes such as *sgc* operon which has been putatively annotated as a sugar uptake and isomerization operon. All 3 strains possessed the *fucA* gene; however, F016 possessed a duplicate allele. Moreover, the F016 strain possessed *scrB* and *scrK* for sucrose metabolism, as well as *mngA* and *mngB*, which are involved in mannose uptake and utilisation.

## ST131 displays reduced selective pressure on select metabolic loci

To explore whether our observations could be applied to a wider selection of genomes, we downloaded assemblies for the ST73 and ST131 lineages from (S3 Table). We screened these assemblies for the 73 genes we selected from our phenotypic analysis, extracted gene sequences, and calculated a Tajima's D value, which is a measurement of selection on a gene with values around 0 indicating an absence of selection. Comparisons between ST73 and ST131 reveal that *treC*, *prpBRDCE*, *cyaY*, *yihU*, *glcB*, *lacA*, and *glnA* have values closer to 0 than ST73, suggesting these alleles are under reduced selective pressures (Fig 4B). Further examination of the genes identified highlighted other differences between ST73 and ST131.

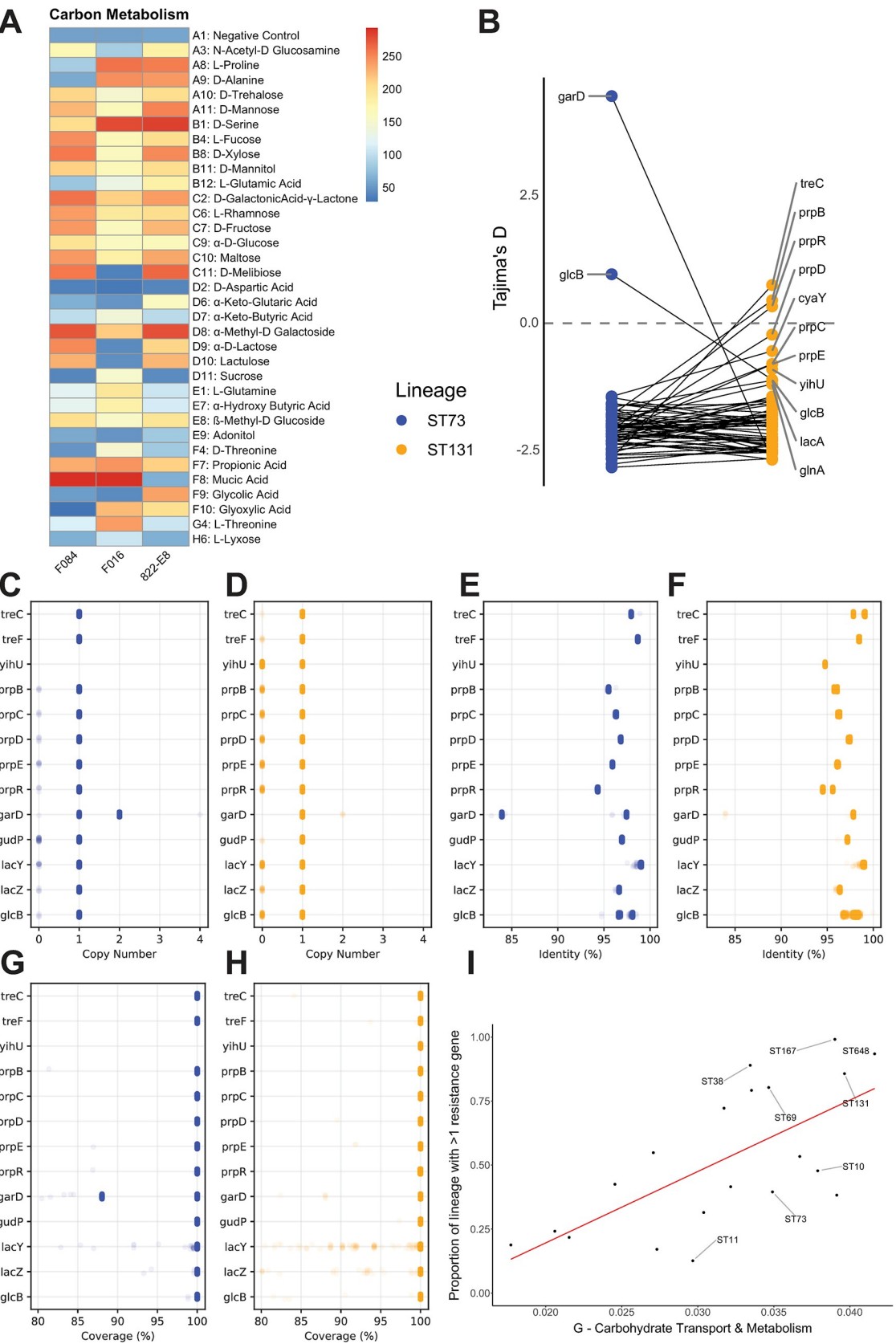

**Fig 4.** (A) Biolog phenotypic microarray analysis of mouse colonising strains. (B) Tajima's D measurements of genes involved in metabolism of differentially utilised metabolites. Genes which display a different value between ST73 (blue) and ST131 (orange) are labelled. (C–H) Genes involved in utilisation of metabolites that show differential patterns between ST73 and ST131 lineages. Number of gene copies per genome in ST73 I and ST131 (D). Percentage identity of K12 reference allele in ST73 (E) and ST131 (F). Percentage coverage of reference allele in ST73 (G) and ST131 (H). (I) The proportion of the accessory pangenome annotated with G COG category (carbohydrate transport and metabolism) against the proportion of each lineage with more than 2 resistance genes. Red line indicates linear regression line (R-squared of 0.4518, *P*-value of 0.0007). Raw data used to produce panel A available in S5 Data. Select metabolic reference gene sequences used to produce panels B–H available at 10.6084/m9.figshare.c.6147189. Genome assemblies and bioinformatics pipeline used to produce panel I are detailed in the Methods section.

Specifically, *yihU* is completely absent from ST73 but present in a significant number of ST131 genomes (Fig 4C and 4D). *LacYZ*, *glcB*, and *prp* operon genes display an elevated level of loss in ST131 (Fig 4C and 4D), with *lacYZ* showing partial gene loss in ST131 (Fig 4 H). While ST73 harbours a partial duplication of *garD* which is linked to the very high Tajima's D value for this gene (Fig 4B, 4C, 4E and 4G). ST131 possess 2 allelic versions of *treC* and *prpR* evident as 2 identity peaks (Fig 4F). *GlcB* appears as 2 allelic variants in ST73 but in ST131, there is more diversity with a broader distribution of identity values (Fig 4E and 4F). Together, this data indicate that ST131 lineage is undergoing a complex evolutionary process targeting metabolic capabilities evident as duplications of core metabolic genes as well as allelic variation.

## MDR lineages display an increase in genetic diversity of carbohydrate metabolism genes in their accessory pangenome associated with recombination of new variant alleles

We sought to examine whether our genomic observations of ST131 were unique or common to multiple MDR lineages of *E. coli*. We curated a dataset of 19,571 *E. coli* genome assemblies encompassing the major *E. coli* phylogroups (A, B1, B2, D, E, and F/G), representing 20 STs incorporating commensal, ExPEC, and EPEC/EHEC lineages (S17A Fig and S3 Table). Antibiotic resistance genes are concentrated in 5 ExPEC lineages: ST38, ST69, ST131, ST167, and ST648 (S17B Fig). These lineages have a high proportion of their population carrying multiple (>1) resistance genes (ST38: 89.0%, ST69: 80.3%, ST131: 85.7%, ST167: 99.1%, ST648: 93.5%). Pangenomes for the 20 different lineages of *E. coli* were constructed with Roary v3.10.2 using an identity threshold of 95%, a core gene frequency of 99% with paralog splitting disabled. This setting allows us to specifically look for unique alleles of core genes as those unique alleles then become part of the accessory genome [13]. The host generalist ST10 had the greatest pangenome size with 46,259 gene clusters identified followed by ST131 with 23,857 clusters (S4 Table and S14 Fig). Core genome size was consistent across all lineages averaging 3,777 genes (S4 Table and S15 Fig). There was no correlation between pangenome size and carriage of AMR genes (S13 Fig). Curiously, AMR carriage was significantly negatively associated with the proportion of phage in the pangenome but showed no correlation with recombination (S17C and S17D Fig). Pangenomes were functionally annotated using the eggNOG database with the eMapper utility, functional composition of the pangenome was explored using Clusters of Orthologous Groups (COG) categories. Links between AMR and biological function were explored in the core and accessory genome using linear regressions analysis that revealed a single significant association. Specifically, lineages with a higher proportion of AMR genes displayed an increased number of "Carbohydrate metabolism and transport" genes in their accessory genome (Fig 4I). There were no other significant correlations between COG categories and carriage of AMR genes after correcting for multiple testing. Our data suggest that diversification of carbohydrate metabolic genes is correlated to the acquisition of multidrug resistance in *E. coli*. We sought to determine the source driving our observed diversity of

carbohydrate genes in AMR lineages. We then looked at the recombination plot created for the main MDR *E. coli* lineage ST131 (S18A Fig) and the main non-MDR ExPEC ST73 (S18B Fig). Our analysis clearly shows recombination occurring in key metabolic loci that does not occur in ST73 and that these loci house metabolism genes occurring as allelic variants in the accessory genome dataset. Together with previously published data, our genomic analysis provides compelling evidence for unique patterns of evolution in metabolism genes within MDR lineages of *E. coli*.

## Discussion

The evolution and global transmission of AMR are a major threat to public health. Genomic identification of antibiotic resistance genes in our global dataset highlights that AMR is concentrated in a number pandemic lineages. This is most evident when examining the average number of resistance genes per genome; ST167, ST648, ST38, and ST131 all display on average in excess of 7 resistance genes per genome. This result is not driven by many resistance genes within a small subpopulation, as ST167, ST648, ST38, and ST131 all have in excess of 80% of their population possessing multiple resistance genes highlighting the success of these lineages. This observation is aligned with numerous other studies that frequently report these populations as major MDR pathogens, confirming previous observations that AMR carriage is not equal across the *E. coli* population with AMR being concentrated in certain lineages [2].

Successful pandemic clones must transmit from the environment to individuals or between individuals rapidly in order to spread globally. Here, we demonstrate that an MDR ST131 strain can readily colonise new hosts even when those hosts are pre-colonised with commensal *E. coli*. The invading ST131 becomes the dominant colonising strain in mice, both when the invading strain is introduced artificially via oral gavage and when mice are co-housed. It is important to emphasise that this transmission is occurring in the absence of any antibiotic treatment conferring a selective advantage to ST131. Typically, it has been required that mice are treated with streptomycin to allow *E. coli* strains to colonise them; however, more recent studies, alongside data presented here, indicate that ST131 strains do not require antibiotic treatment in order to competitively colonise mice [17]. In our study, the ST131 out-colonises another ExPEC strain of the common ST73 lineage. This implies that ST131 possesses some mechanism by which it can out-compete commensals that is lacking in another highly successful but non-MDR ExPEC lineage ST73. While our study is limited to the use of germ-free mice, our observations mimic those made from human traveller studies which have observed MDR *E. coli* as frequent colonisers of healthy travellers in the absence of antibiotics [7–9]. Within household transmission has also been observed for ST131 [18]. Studies of healthy individuals who travel to regions where antibiotic resistance is endemic have reported varying levels of colonisation of between 30% and 70% upon return [7]. Sampling travellers during their trip revealed a much higher rate of colonisation rate of 95%, of which *E. coli* was the most common colonising bacteria [9]. From our data, the resident commensal strain was not completely displaced, similar observations have been made of human travellers, specifically individuals colonised by MDR *E. coli* had a recurrence of their original commensal *E. coli* at the end of their travels [8].

Phenotypic microarray of our colonising strains again pointed to altered utilisation of carbon sources. Subsequent targeted genomic analysis highlighted altered mutational profiles of our assayed strains. We expanded this analysis to multiple MDR lineages of *E. coli* in comparison to multiple non-MDR ExPEC lineages revealing diversification of carbohydrate metabolic genes in multiple MDR lineages. The genomic signature we have identified is complex and requires further investigation. Previous pangenome analysis identified metabolic loci as being

enriched in nucleotide diversity in ST131 compared to other ExPEC, specifically anaerobic metabolic genes were exhibiting increased genetic variation [13]. Our functional pangenome analysis supports these observations, revealing that there is a significant correlation between carriage of AMR genes and genetic diversity in metabolism genes. Specifically, there is increased variation in genes encoding carbohydrate metabolism in lineages with a high rate of antibiotic resistance carriage. Previous analyses have focussed on individual lineages, whereas here, we present data on multiple MDR ExPEC lineages, revealing that metabolic variation is a shared adaptation of MDR *E. coli*. Experimental evolution studies have identified that *E. coli* can evolve resistance to antibiotic stress through mutations in core metabolic genes, particularly those involved in carbon and energy metabolism [19]. These observed mutations occur at low frequency and were only detected through sequencing of multiple isolates from a population; however, they were still detectable in datasets of clinical samples demonstrating their relevance. A large-scale bacterial genetic screen to test the effect of allelic variation in key metabolism genes on the ability to colonise and displace commensal *E. coli* in the mammalian intestinal tract would seem attractive. However, the signal observed in our dataset occurs in multiple genes and pathways and genetically investigating such a polygenic trait is far from trivial. We suggest further investigation of our findings will need to combine classical genetics with long-term and complex experimental studies attempting to recapitulate MDR clone evolution.

Collectively, our data demonstrate that MDR *E. coli* is highly capable of host intestinal colonisation, displacing resident commensal *E. coli* to become the dominant strain and readily transmitting between hosts. Our genomic analysis implicates metabolism as a pivotal factor in the evolution of AMR linked to the incredible gut colonisation ability of MDR *E. coli*.

## Methods

### Ethics statement

Animal protocols were reviewed and approved by the University of Calgary Animal Care Committee (approved protocol numbers AC17-0090 and AC19-0139) and animal experiments were conducted in accordance with Canadian Council on Animal Care guidelines.

### Mouse colonisation

Mouse colonisation experiments were conducted at the International Microbiome Centre, University of Calgary, Canada. Germ-free mice were bred and maintained in flexible film isolators in our axenic breeding facility, and germ-free status was confirmed by a combination of Gram staining, Sytox green DNA staining, anaerobic and aerobic culture, and 16S rRNA gene amplicon sequencing from faeces were maintained in isocages in our gnotobiotic facility. Germ-free C57BL/6 mice were colonised with $10^9$ CFU of bacteria via oral gavage (see S3 Table and S4 and S5 Figs for details of strains used for colonisation) and maintained in sterile isocages in our gnotobiotic facility throughout the duration of experiments. Bacterial colonisation was monitored by CFU enumeration on UTI Chromogenic Agar (Thermo) from faecal pellets as well as by DNA extraction and strain-specific probe-based qPCR. DNA from faecal pellets was extracted using the MagMAX Microbiome Ultra Nucleic Acid isolation kit (Thermo) on a KingFisher Flex instrument (Thermo) following manufacturer's instructions. Strain-specific primers and probes were designed to target unique genes (822-E8: *clpP*, F084: *lon*, F016: *prtR*–S4 Table) identified from genome data. Probes were manufactured by Integrated DNA Technologies (IDT). Reactions were performed using PrimeTime Gene Expression master mix (IDT) on a QuantStudio 1 system (Thermo). Reactions were performed following manufacturer's recommended parameters: 3 min at 95*C, 40 cycles of 15 s at 95*C, 1

min at 60*C, fluorescence readings taken at the end of the extension stage. A standard curve was generated from a bacterial culture of known CFU.

## In vitro supernatant cultures

Bacteria were grown in LB broth to an OD600 of between 0.4 and 0.6, cultures were pelleted at 4,000 rpm for 15 min. The supernatant was passed through a 0.22 μm filter, the filtrate was diluted in a 1 to 1 ratio with fresh LB. Bacteria were inoculated into the supernatant mixture and grown in a 96-well plate at 37˚C with growth measured by OD600 readings at 10-min intervals by a Spark Microplate Reader (Tecan).

## Functional metabolic analysis

Strains (F084, F016, and 822 E8) were grown on LB-NaCl agar (5 g/L Yeast Extract (Melford, Y20020-500.0), tryptone 10 g/L (Melford, T60060-500.0), and agar (Melford, A20020-500.0)) for 16 h. Inoculations were set up following manufacturer's instructions with below modifications. Biomass was removed with a cell scraper and suspended in inoculation fluid 0a (IF-0a – Techno-path, 72268—PM IF-0a GN/GP Base (1.2×) 125 ml) to an O.D.600 of 0.185 +/− 0.05. This was then diluted 1:6 with IF-0a containing 1.4% (v/v) of Biolog Redox Dye Mix A (100×, Techno-path, 74221). To each well of each PM1 plate (Techno-path, 12111), 100 μl of this suspension was added and strains incubated for up to 48 h at 37˚C, static, in a OmniLog PM system (imaging every 15 min). Data was extracted using Biolog softwares (conversion of D5E to OKA: D5E_OKA Data File Converter v1.1.1.15 and extraction of raw kinetic data using PM analysis software: Kinetic V1.3). Statistically relevant results were identified using a one-way ANOVA with a $P$-value threshold of 0.05.

## Genomic dataset curation, pangenome construction, and AMR gene detection

A total of 20 lineages or sequence types (STs) were selected from the literature with a focus on ExPEC lineages but also including EHEC and EPEC clones. This resulted in a dataset of 19,571 *E. coli* genome sequences encompassing all the *E. coli* phylogroups (A, B1, B2, D, E, and F/G) (S1 and S2 Tables) [10.6084/m9.figshare.c.6147189]. The earliest samples with reliable metadata were from 1980; however, the majority was sequenced in recent decades. Humans represented the major source niche for all lineages except ST117 for which poultry was the major niche. This dataset contained samples from multiple countries; however, Europe and North America accounted for the majority.

Genome assemblies for each lineage were downloaded from Enterobase [20] using a custom python script [https://github.com/C-Connor/EnterobaseGenomeAssemblyDownload]. Duplicated assemblies were identified using Mash v1.1.1 [21] to estimate genome similarity, a custom R script then removed isolates with a Mash distance of 0 [https://github.com/C-Connor/MashDistDeReplication]. Dendrograms of Mash distances were also constructed and examined for outlier genomes that were not part of a larger cluster. The remaining genome files were annotated with Prokka v1.12 [22] and pangenomes were constructed with Roary v3.10.2 [23] using a 95% identity threshold, a 99% core genome threshold, paralog splitting was disabled, and a core genome alignment was produced using MAFFT. AMR genes were detected using Abricate v 0.8 [https://github.com/tseemann/abricate] with the Resfinder-2018 database [24], results were filtered to remove hits with less than 80% resistance gene coverage. A phylogeny of the whole dataset was constructed using MashTree v0.36.2 [25] and visualised in iTOL [26].

## Targeted genomic comparisons

Genes of interest were selected based on their involvement in the utilisation of Biolog metabolites. Short read data for F084 and F016 strains was mapped to the commensal 822-E8 strain using Snippy 4.6.0 (https://github.com/tseemann/snippy). Reference gene sequences from (K12 MG1655 U00096.3) were downloaded from NCBI and genome assemblies were screened using Abricate. Gene seqeuences were extracted from each assembly using extract_genes_ABRricate.py (https://github.com/boasvdp/extract_genes_ABRicate). Sequences were aligned with MAFFT 7.487 using default parameters. Tajima's D measurements from the alignments were calculated in R using the packages ape 5.7.1 [39] and pegas 1.2

## Pangenome functional annotation

Pangenome reference Fasta files produced by Roary were functionally annotated using emapper-1.0.3-3-g3e22728 [27] based on eggNOG orthology data [28]. Sequence searches were performed using DIAMOND [29]. Functional annotation data was combined with the gene presence absence matrix and analysed in R v4.0.3. To examine if there was any association between functional composition of the accessory pangenome or core genome linear regression was performed between carriage of AMR (as a proportion of the population with 2 or more AMR genes) and individual COG categories, correcting for multiple testing.

## Supporting information

**S1 Method. Supplementary methodology for histological and cytokine analysis of mouse tissues.**
(DOCX)

**S1 Table. Details of *E. coli* strains used for mouse colonisation and Biolog experiments.**
(XLSX)

**S2 Table. Primers and probes used for CFU quantification qPCR and cytokine qPCR.**
(XLSX)

**S3 Table. Numbers of *E. coli* genomes from Enterobase analysed.**
(XLSX)

**S4 Table. Pangenome size of each *E. coli* lineage analysed.**
(XLSX)

**S1 Fig. Heatmap of virulence factors identified in mouse colonising strains.**
(EPS)

**S2 Fig. Heatmap of AMR genes identified in mouse colonising strains.**
(EPS)

**S3 Fig. CFU of colonising bacteria in monocolonised mice measured by enumeration from UTI Chromogenic agar plates.** CFU counts were adjusted to faecal pellet weight. Geometric average CFU of 822-E8 (black line), F084 (blue line), and F016 (orange line) is shown with geometric standard deviation, 3 replicates for each condition. Raw data used to produce figure available in S6 Data.
(EPS)

**S4 Fig. Inflammatory histology score for sections of small intestine and colon from colonised mice and germ-free control mice.** Small intestine histology scores for monocolonised mice (A), co-incolucated/competitive colonisation mice (B), and monocolonised mice

challenged with a second strain after 1 week (C). Colon section histology scores for monocolonised mice (D), co-incoluated/competitive colonisation mice (E), and monocolonised mice challenged with a second strain after 1 week (F). Scores were assigned to inflammatory cell infiltrate (black circles), changes to epithelium (green squares) and overall mucosal architecture (blue triangles). Score of 1 indicates homeostatic mucosa while a score of 5 indicates severe inflammatory disruption. Points represent scores for individual fields of view (FoV) taken from multiple tissue sections. Raw data used to produce figure available in S7 Data.
(EPS)

**S5 Fig. Representative histological sections of colonised mouse small intestine and colon stained with HE, 100 μm scale bar.** Germ-free control mice are included.
(EPS)

**S6 Fig.** Cytokine expression in the small intestine (A), caecum (B), and colon (C) of monocolonised mice 3 weeks after colonisation. Mice were colonised with a 822-E8 (black), F084 (blue), or F016 (orange). Germ-free mice were also assayed (open circles). Expression of cytokines is displayed as the delta Ct between cytokine and housekeeping gene *Pol2ra*, negative values indicate lower expression than housekeeping gene. Statistical significance determined with 2-way ANOVA with Tukey's multiple comparisons, $n = 3$ for all conditions. Raw data used to produce figure available in S8 Data.
(EPS)

**S7 Fig.** Cytokine expression in the small intestine (A), caecum (B), and colon (C) of co-inoculated mice 1 week after colonisation. Mice were colonised with an 822-E8 and F084 (blue, $n = 5$), 822-E8 and F016 (orange, $n = 4$), or F084 and F016 (purple, $n = 5$). Germ-free mice were also assayed (open circles, $n = 3$). Expression of cytokines is displayed as the delta Ct between cytokine and housekeeping gene *Pol2ra*, negative values indicate lower expression than housekeeping gene. Statistical significance determined with 2-way ANOVA with Tukey's multiple comparisons. Raw data used to produce figure available in S9 Data.
(EPS)

**S8 Fig.** Cytokine expression in the small intestine (A), caecum (B), and colon (C) of monocolonised mice which were challenged with a second strain 7 days after initial colonisation. Mice were colonised with 822-E8 and challenged with PBS (pink, $n = 5$), 822-E8 challenged with F084 (blue, $n = 5$), 822-E8 challenged with F016 (orange, $n = 4$), or F016 challenged with 822-E8 (black, $n = 4$). Germ-free mice were also assayed (open circles, $n = 3$). Expression of cytokines is displayed as the delta Ct between cytokine and housekeeping gene *Pol2ra*, negative values indicate lower expression than housekeeping gene. Statistical significance determined with 2-way ANOVA with Tukey's multiple comparisons. Raw data used to produce figure available in S10 Data.
(EPS)

**S9 Fig. Cycle threshold values for strain-specific probes assayed using DNA extracted from faecal pellets from monocolonised mice; 822-E8 specific probe (clpP) in blue, F084 specific probe (lon) in green, and F016 specific probe (prtR) in orange.** Raw data used to produce figure available in S11 Data.
(EPS)

**S10 Fig. Standard curves for CFU determination.** DNA isolates from broth culture of known CFU serially 10-fold diluted and plotted against Ct for that dilution. Colour lines indicate standard curve from stock dilutions, while black lines indicate standard curves from every assay plate used for CFU quantitation. Standard curves for 822-E8 specific probe clpP (A), F084

probe lon (B), and F016 probe prtR (C). Raw data used to produce figure available in S12 Data.
(EPS)

**S11 Fig.** Source niche for genome in dataset (A) and cumulative time of collection (B). Metadata used to produce figure available at 10.6084/m9.figshare.c.6147189.
(EPS)

**S12 Fig. Geographic distribution of genomes in dataset, colour scale is logarithmic.**
Figure was produced in R using ggplot2 (https://ggplot2.tidyverse.org) with a map layer from
Natural Earth (https://www.naturalearthdata.com/). Metadata used to produce figure available
at 10.6084/m9.figshare.c.6147189.
(EPS)

**S13 Fig.** Correlation between pangenome size and average number of resistance genes (A).
Correlation between pangenome size and proportion of the lineage with 2 or more resistance
genes (B).
(EPS)

**S14 Fig. Pangenome rarefaction curves for the accessory compartment of each ST pangenome.**
(EPS)

**S15 Fig. Core genome rarefaction curves for each ST.**
(EPS)

**S16 Fig.** Pangenome total size against the proportion of genes annotated as hypothetical proteins for each ST (A). Pangenome total size against the proportion of the pangenome assigned
a COG category for each ST (B).
(EPS)

**S17 Fig. Phylogenetic tree of selected lineages of *E. coli*, incidence of AMR genes within the
dataset and correlations between HGT and AMR gene content.** (A) Phylogenetic tree of
dataset produced from Mash distances, tip branches are coloured by ST and phylogroups are
outlined. (B) The proportion of the lineage with 2 or more resistance genes of any class against
the proportion of the lineage with 1 or more beta-lactamase gene. (C) Proportion of the pangenome that is associated with phage elements against the proportion of the lineage that has 2
or more resistance genes of any class. (D) Proportion of the genome predicted to be within a
recombination block against the proportion of the lineage that has 2 or more resistance genes
of any class. Red lines indicate linear regressions (B: $R^2$ 0.954, *P*-value $1.61 \times 10^{-13}$ C: $R^2$ 0.369,
*P*-value 0.0045, D: $R^2$ 0.138, *P*-value 0.117).
(EPS)

**S18 Fig.** Recombination regions predicted by Gubbins in ST73 (A) and ST131 (B) lineages.
Red blocks indicate putative regions of recombination. Yellow and orange track bar indicate
position in reference genome. Select regions are annotated.
(PDF)

**S1 Data. Colony forming units from co-inoculated mice as measured by ST-specific qPCR.**
Data used to produce Fig 1B–1E.
(XLSX)

**S2 Data. Growth curves of strains used to colonise mouse in culture supernatant from
competing strains.** Data used to produce Fig 1F–1H.
(XLSX)

**S3 Data. Colony forming units from colonised mice with subsequent challenge as measured by ST-specific qPCR.** Data used to produce Fig 2B–2F.
(XLSX)

**S4 Data. Colony forming units from co-housed mice with as measured by ST-specific qPCR.** Data used to produce Fig 3B–3D.
(XLSX)

**S5 Data. Biolog phenotypic microarray data for plate PM1.** Data used to produce Fig 4A.
(XLSX)

**S6 Data. Colony forming units from mice colonised by a single strain measured by plating.** Data used to produce S3 Fig.
(XLSX)

**S7 Data. Inflammatory scoring for histological sections of mouse small intestine and colon.** Data used to produce S4 Fig.
(XLSX)

**S8 Data. Expression of a panel of cytokines in the small intestine, caecum, or colon of mice colonised by a single strains, measured by qPCR.** Data used to produce S6 Fig.
(XLSX)

**S9 Data. Expression of a panel of cytokines in the small intestine, caecum, or colon of mice co-inoculated with 2 strains, measured by qPCR.** Data used to produce S7 Fig.
(XLSX)

**S10 Data. Expression of a panel of cytokines in the small intestine, caecum, or colon of mice pre-colonised with subsequent challenge, measured by qPCR.** Data used to produce S8 Fig.
(XLSX)

**S11 Data. Cross reactivity of ST-specific probes.** Data used to produce S9 Fig.
(XLSX)

**S12 Data. Standard curves for ST-specific probes used to quantify CFU.** Data used to produce S10 Fig.
(XLSX)

## Author Contributions

**Conceptualization:** Jukka Corander, Alan McNally.

**Data curation:** Christopher H. Connor.

**Formal analysis:** Christopher H. Connor, Amanda Z. Zucoloto, John T. Munnoch, Paul A. Hoskisson, Alan McNally.

**Funding acquisition:** Christopher H. Connor, Paul A. Hoskisson, Alan McNally.

**Investigation:** Christopher H. Connor, John T. Munnoch, Ian-Ling Yu, Paul A. Hoskisson, Braedon McDonald, Alan McNally.

**Methodology:** Christopher H. Connor, Amanda Z. Zucoloto, John T. Munnoch, Ian-Ling Yu, Jukka Corander, Braedon McDonald, Alan McNally.

**Project administration:** Christopher H. Connor, Paul A. Hoskisson, Braedon McDonald, Alan McNally.

**Resources:** Paul A. Hoskisson, Braedon McDonald, Alan McNally.

**Software:** Jukka Corander.

**Supervision:** Amanda Z. Zucoloto, Paul A. Hoskisson, Braedon McDonald, Alan McNally.

**Visualization:** Christopher H. Connor, John T. Munnoch.

**Writing – original draft:** Christopher H. Connor, Jukka Corander, Alan McNally.

**Writing – review & editing:** Christopher H. Connor, Amanda Z. Zucoloto, John T. Munnoch, Ian-Ling Yu, Jukka Corander, Paul A. Hoskisson, Braedon McDonald, Alan McNally.

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
