## [Editor Report · Decision Letter 0]

2 Dec 2022

Dear Alan, 

Thank you for submitting your manuscript entitled "Multi-drug resistant E. coli displace commensal E. coli from the intestinal tract, a trait associated with elevated levels of genetic diversity in carbohydrate metabolism genes" for consideration by PLOS Biology.

As previously discussed, we would be happy to send it for peer-review as a Discovery Report (please choose this article type when finalising your submission). 

Before we can send your manuscript to reviewers, we need you to complete your submission by providing the metadata that is required for full assessment. To this end, please login to Editorial Manager where you will find the paper in the 'Submissions Needing Revisions' folder on your homepage. Please click 'Revise Submission' from the Action Links and complete all additional questions in the submission questionnaire.

Once your full submission is complete, your paper will undergo a series of checks in preparation for peer review. After your manuscript has passed the checks it will be sent out for review. To provide the metadata for your submission, please Login to Editorial Manager (https://www.editorialmanager.com/pbiology) within two working days, i.e. by Dec 04 2022 11:59PM.

Kind regards,

Nonia

Nonia Pariente, PhD

Editor in Chief

PLOS Biology

npariente@plos.org

---

## [Decision Letter · Decision Letter 1]

6 Feb 2023

Dear Dr. McNally,

Thank you for your patience while your manuscript "Multi-drug resistant E. coli displace commensal E. coli from the intestinal tract, a trait associated with elevated levels of genetic diversity in carbohydrate metabolism genes." was peer-reviewed at PLOS Biology. It has now been evaluated by the PLOS Biology editors, an Academic Editor with relevant expertise, and by several independent reviewers. 

In light of the reviews, which you will find at the end of this email, we would like to invite you to revise the work to thoroughly address the reviewers' reports.

As you will see below, all the reviewers find the manuscript interesting. However, they all think that your claims are too general or your conclusions regarding the carbohydrate metabolism are not totally convincing for a Discovery Report format. The reviewers ask to expand this point and add more mechanistic insights. We can consider the manuscript as a Research Article and we think that the manuscript will improve if you add cause-and-effect experiments and mechanistic insights.

Given the extent of revision needed, we cannot make a decision about publication until we have seen the revised manuscript and your response to the reviewers' comments. Your revised manuscript is likely to be sent for further evaluation by all or a subset of the reviewers.

**IMPORTANT - SUBMITTING YOUR REVISION**

*Re-submission Checklist*

*Published Peer Review*

*PLOS Data Policy*

*Blot and Gel Data Policy*

Sincerely,

Paula

---

Senior Editor

PLOS Biology

REVIEWS:

Reviewer #1: Microbiome.

Reviewer #2: Maria Hadjifrangiskou. E. coli biology and host interaction.

Reviewer #3: Bacterial growth and interactions.

Reviewer #1: In the manuscript „Multi-drug resistant E. coli displace commensal E. coli from the intestinal tract, a trait associated with elevated levels of genetic diversity in carbohydrate metabolism genes" Connor and colleagues combine colonization experiments in germfree mice and bioinformatical analyses to study niche competition between three E. coli strains in the gut. They demonstrate that the tested MDR ST131 strain can reduce colonization levels of a non-MDR ST73 strain by approximately a factor of 5 in absence of antibiotic selection and in different colonization settings. Bioinformatical analysis performed by the authors show an association between AMR genes and genes involved in carbohydrate utilization in ExPEC, which leads the authors to suggest that the displacement of commensal E. colis from the gut by MDR E. coli ST131 could be mediated by carbohydrate competition and a fitness advantage of MDR E. coli strains.

Major comments: (specific comments follow below)

1. While the utilized germfree model is very simplistic and allows in vivo comparison of direct competition, I have my doubts how general the conclusions are in light of recent studies showing that commensal bacteria beyond the family Enterobacteriaceae contribute to competition between related bacteria in the gut (Eberl et al, Cell Host Microbe 2021 and Osbelt et al. Cell Host Microbe 2021). Moreover, in germfree mice, many colonization relevant carbohydrates are not released from dietary polysaccharides.

2. Very little evidence is provided that competition for carbohydrates and not differences in the ability to acquire other growth limiting factors such as amino acids and trace elements or the ability to utilize redox acceptors contribute to the competition in vivo. Therefore, how are the two analysis linked to each other?

3. The hypothesis of the authors are highly relevant, but they generalize their observations too much. Single strains are tested, which limits the conclusions that can be drawn. I would suggest that the authors should make this much clearer by referring to strains rather the STs throughout the different sections of the manuscript, including the abstract, results, and discussion.

Specific comments:

Line 142: should be "16S rRNA" There is also a sentence fragment in that line.

Line 227ff: Make sure to refer to specific strains rather than STs. In general, throughout the whole study, just one strain was used per condition (MDR, non-MDR, commensal). To see whether the effect is strain specific or if it's a general phenotype that MDR ExPEC strains outcompete commensal strains, experiments should be repeated with different strains of each category.

Line 229ff: In general the authors do not give a comprehensive description of the strains… Also, if carbohydrate utilization is the key factor, why do not test growth in different carbon sources for those three strains? What are the differences in carbohydrate preferences? Can they equally grow in important sugars available in the gut of mice (e.g sucrose, glucose, cellobiose etc.). The statement of "carbohydrate utilization" is subsequently very broad and not really followed on in the whole study? By growing strains in different media and monitoring the general carbohydrate utilization capability, it would be at least possible to identify a group of potential candidate sugars. 

Line 239f: Data shown in FigS3 shows the colonization levels of a mono-colonization with the strains. This data shows, that the MDR ExPEC strain shows 5-10 fold higher colonization levels in germfree mice at day 1-5. This could influence the following phenotypes. Could strains with similar colonization levels be used? If no extra experiments are performed, please discuss this limitation. 

Line 241ff: It is interesting that apparently the strains do replace each other in the gut. What happens in a defined community of strains? It is rather unlikely that travelers harbor no community in the gut when acquiring new E. coli strains. Is this working in a similar manner, when mice with low or high complex communities are used? E.g. are the E. coli strains naturally colonizing the mice from different vendors also replaced by the superior E. coli strains?

Line 248 ff. Are the strains predicted to be capable to produce toxins and phages? If so, is LB a sufficient medium to induce lytic prophage induction and release of toxins? In other enterobacterial species, toxin production is tightly regulated (e.g induction for specific substrates like sugar or iron). I would suggest, to include other modified media to exclude that toxins or phages are relevant. 

If the MDR strains encode lytic prophages, it would be a clearer experiment to generate phage deficient strains and test, whether those strains fail to replace the commensal strain in vivo. Also, plaque-formation in the commensal bacteria with intestinal content of MDR-colonized could be evaluated to see whether in vivo something different happens.

In a similar direction: what happens, if strains are co-cultured in minimal media with different carbon sources? Do strains co-exist or is the commensal strain equally outcompeted? It would be at least crucial to prove in simplified in vitro experiments that the strains do outcompete each other for different substrates.

Line 293ff: : The data are largely descriptive, and none of the observations is followed on mechanistically. The mice used in this study are raised germfree, so its absolutely obvious that challenge with any kind of bacteria will stimulate the immune system. Different cytokines were measured to "assess host mucosal immune response" and the author claim relatively few differences between colonization conditions. Does that mean, that the immune response is dispensable? I don't understand, why this data are included in the manuscript. If the focus is shifted to potential differences in the immune system, the authors should provide more detailed data than one histology picture and levels of 11 cytokines. 

Specifically, line 307ff "Collectively, these data demonstrate that displacement of commensal E. coli and dominant colonization by MDR ExPEC is associated with a distinct host mucosal immune response." 

I think, the few data on cytokine profiles do not support a statement like this…

Line 355ff: The authors identify an "increased number of carbohydrate metabolism and transport" genes. 

Does this translate in an increased ability of the MDR strains to utilize different carbohydrates? I suggest analyzing the three isolates using the Biolog Screening System to verify that the MDR strains indeed utilize a more diverse set of carbohydrates.

Where specific carbohydrates identified as being most prominent in MDR strains? Are these carbohydrate utilization genes relevant in the mouse intestine? 

What happens when carbohydrate levels are increased in the mouse gut? If the carbohydrate utilization is the major factor of intraspecies competition, an increased amount of carbohydrates available in the mouse gut should impact the phenotype. This could be done by a supplementation with specific csrbohydrates, e.g., ones that are well/solely utilized by the commensal strains.

Reviewer #2: This is a compelling study that begins to investigate how ExPEC MDR isolates behave in the host compared to non-MDR isolates. The study is significant because it begins to add a much needed layer of better understanding the physiology of these pathogens as opposed to merely looking at genomic content. 

I have a few questions/comments for the authors to consider:

1) The concept of altered metabolism is extremely interesting, but not discussed or expanded at all in the current study. This results section should be expanded and validated. Are there specific metabolism gene in CHO metabolism that are different? Are there more of them? What if a subset was transplanted to a non-MDR ExPEC strain? I may have missed this, but if the analysis was confined to the ExPECs, would this difference in metabolic gene carriage still be the same. 

2) Gnotobiotic mouse experiment: If the MDR strain is cured (to create an isogenic non-MDR clone), would the advantage persist? These studies are elegant, but may be worth repeating in a conventionally reared mouse strain that lacks Enterobacteriaceae to observe the interaction of the strains with the rest of the microbiota. finally, given the emphasis on ExPEC, the authors should evaluate transition of the strains to the bladder by performing longitudinal urinalysis. It would be very interesting to see if the MDR strain outcompetes in transmission as well. 

Reviewer #3: In the manuscript by Connor et al., the authors use a gnotobiotic mouse model to demonstrate that an MDR E. coli strain is capable of out-competing and also of displacing non-MDR E. coli from the gut. This result is interesting since it does not require antibiotics, indicating that the MDR strain has some adaptation that increases its fitness. This part of the manuscript is generally convincing, although I do have some questions that I outline below. However, the rest of the manuscript seems preliminary, because they rely on a bioinformatic analysis that is not particularly convincing - it seems to rest on a single correlation with increased diversity of carbohydrate metabolism genes, when they have yet to prove that the fitness results are general and also they do not pursue any of the hypotheses of this correlation. Thus, while I found the initial result intriguing, the paper did little to dig into the mechanism underlying this observation, and hence the observation itself seems undercapitalized on. Indeed, the manuscript suffers throughout from the lack of follow up or misinterpretation of observations. Moreover, the metabolic hypothesis is not consistent with the rest of their data, as I discuss below.

Major comments:

- Line 246: I was surprised that the ExPEC strains coexisted for 6 days, since I assume there are substantial differences. The authors should discuss the genomic differences between these two strains.

- Line 253: the growth curves in 1F-H do not rule out toxins or phage. Phage in particular are likely to be produced in times of stress that may not be mimicked by the in vitro growth environment but are present in the gut. Thus, more experiments are needed or this statement needs to be toned down a lot.

- Why could the non-MDR ExPEC strain not displace? This also seems like a worthy subject of at least discussion based on genomic content, and provides the substrate to generate candidates for knockout study in the MDR strain.

- Line 288: I don't see why the CFU equivalency means growth rate is the same.

- Line 303: I don't understand why the hypothesis of carbohydrate metabolism would lead to the prediction of cytokine response and inflammation, as they observe?

- Moreover, why is the inflammatory response only with gavage? Seems like it is something to do with initial colonization and adaptation. And if it only appears with gavage, seems like it is much less relevant to the typical human scenario, since we are not really gavaged with large quantities of pathogen.

- Line 336: the statement that AMR lineages are more recombinogenic but to the exclusion of phage seems to be implying much more mechanism than the correlation merits. Could easily be an indirect correlation. Also I don't see why it would be the proportion that matters - likely there is one or a few genes that enables metabolism of particular aspects of the diet that causes the fitness advantage in their strain.

- Most of all, while I find the association of fitness with carbohydrate metabolism interesting, it is in some sense expected since previous evolution experiments in mice have shown that E. coli evolves by selecting for carb metabolism genes. Moreover, basing it on the correlation in Fig. 6 is highly speculative. For instance, they have only 3 strains from which to base their fitness idea in the first place - it could be that other MDR strains do not displace commensal E. coli!

- It would likely be straightforward to make a transposon library in this strain, which could then be screened for growth fitness, given the hypothesis of metabolism. This would provide a mechanistic test of their idea.

- 

Other comments:

- The whole paragraph in the introduction ending on line 100 about recent studies of E. coli colonization seems like a series of anecdotes that could just be summarized in a single sentence, rather than going through all of the details.

- Line 243: it would be interesting to provide some estimates of the fitness advantage of the ExPEC strains.

- Throughout the manuscript in many places, the authors are not careful about verb tense. Many times things should be in past tense rather than the present that is in the manuscript - e.g. line 263.

- Line 333: why "incredibly"?

---

## [Decision Letter · Decision Letter 2]

24 Aug 2023

Dear Dr. McNally,

Thank you for your patience while we considered your revised manuscript "Multi-drug resistant E. coli displace commensal E. coli from the intestinal tract, a trait associated with elevated levels of genetic diversity in carbohydrate metabolism genes." for publication as a Discovery Report at PLOS Biology. This revised version of your manuscript has been evaluated by the PLOS Biology editors, the Academic Editor, and the original reviewers.

Based on the reviews and on our Academic Editor's assessment of your revision, we are likely to accept this manuscript for publication as a Discovery Report, provided you satisfactorily address the following data and other policy-related requests.

Please note that the reviewers are not completely satisfied with the revision as the new data added still does not establish cause-and-effect in a more relevant model. Therefore, we will consider this manuscript only as a Discovery Report. For that, you need to reduce the number of main figures to a maximum of 4.

Discovery Reports describe novel and intriguing initial findings with the potential to lead to a significant new result for the field. Discovery Reports are short articles, typically with 2-4 main figures. While the research may be preliminary, studies should be advanced to the stage where observations or findings have been confirmed by independent methods or experimental approaches and obvious alternative interpretations have been ruled out. Discovery Reports are designed to work together with Update Articles to empower researchers to evaluate and share work in a way that more closely mirrors the real-world research process and create a comprehensive research story.

1. ETHICS STATEMENT:

-- Please include the full name of the IACUC/ethics committee that reviewed and approved the animal care and use protocol/permit/project license. Please also include an approval number.

-- Please include the specific national or international regulations/guidelines to which your animal care and use protocol adhered. Please note that institutional or accreditation organization guidelines (such as AAALAC) do not meet this requirement.

2. DATA POLICY:

A) Supplementary files (e.g., excel). Please ensure that all data files are uploaded as 'Supporting Information' and are invariably referred to (in the manuscript, figure legends, and the Description field when uploading your files) using the following format verbatim: S1 Data, S2 Data, etc. Multiple panels of a single or even several figures can be included as multiple sheets in one excel file that is saved using exactly the following convention: S1_Data.xlsx (using an underscore).

B) Deposition in a publicly available repository. Please also provide the accession code or a reviewer link so that we may view your data before publication.

Regardless of the method selected, please ensure that you provide the individual numerical values that underlie the summary data displayed in the following figure panels as they are essential for readers to assess your analysis and to reproduce it: Figures 1BCDEFGH, 2BCDEF, 3BCD, 4ABCDEFGH, 5ABCD, 6ABC, and Supplementary Figures S3, S4ABCDEF, S6ABC, S7ABC, S8ABC, S9, S10, S11AB.

**Please also ensure that figure legends in your manuscript include information on where the underlying data can be found, and ensure your supplemental data file/s has a legend.**

3. Please add size bars to microscopy pictures in Figure S5.

4. Please move the Materials and Methods to the end of the manuscript, after the discussion.

5. We suggest a change in the title: " Multi-drug resistant E. coli encoding high genetic diversity in carbohydrate metabolism genes displace commensal E. coli from the intestinal tract."

We expect to receive your revised manuscript within two weeks.

*Published Peer Review History*

*Press*

Sincerely,

Paula

---

Senior Editor,

pjaureguionieva@plos.org,

PLOS Biology

Reviewer remarks:

Reviewer #1: The authors have addressed our questions and comments reasonably and we congratulate them to this study.

Reviewer #2: The authors have not adequately responded to all the reviewer's comments, with most of the responses "extending beyond the current study".

The authors present in vitro Biolog data for 3 strains but do not expand their in vivo work to more than one MDR strain. In the very least, they should perform an experiment to see if they challenge F06 with another MDR strain do they both colonize?

How about if they challenge with another B2 UPEC that is non-MDR? Studies have shown that UPEC stays in the gut displacing commensals over long periods of time in humans (See Forde et al, from Mulvey and Schembri's groups). It would be most beneficial if the authors actually performed this key experiment.

Reviewer #3: The authors have done some work to address the reviewer feedback including Biology measurements and a comparative genomics analysis. I was disappointed not to see more of an effort to address the more preliminary aspects of the article, in particular efforts to construct a transposon library, which is effective in many E. coli strains (or at least a report of low transformation efficiency of their strains). The new additions have added impact to the paper.

I leave it to the editor as to the decision of the general interest in this topic.

---

## [Editor Report · Decision Letter 3]

11 Sep 2023

Dear Dr. McNally,

Thank you for the submission of your revised Discovery Report "Multi-drug resistant E. coli encoding high genetic diversity in carbohydrate metabolism genes displace commensal E. coli from the intestinal tract." for publication in PLOS Biology. On behalf of my colleagues and the Academic Editor, Sebastian Winter, I am pleased to say that we can in principle accept your manuscript for publication, provided you address any remaining formatting and reporting issues. These will be detailed in an email you should receive within 2-3 business days from our colleagues in the journal operations team; no action is required from you until then. Please note that we will not be able to formally accept your manuscript and schedule it for publication until you have completed any requested changes.

PRESS

Sincerely, 

Paula

---

Senior Editor

PLOS Biology
